

# Mineral physical protection and carbon stabilization in-situ evidence
# revealed by nano scale 3-D tomography
Yi-Tse Weng[1,#], Chun-Chieh Wang[2,#], Cheng-Cheng Chiang[2], Heng Tsai[3], Yen-Fang
Song[2], Shiuh-Tsuen Huang[4], Biqing Liang[1,*]
[1]National Cheng Kung University, Department of Earth Sciences, Tainan, Taiwan ROC
[2]National Synchrotron Resource Research Center, Hsinchu, Taiwan ROC
[3]National Changhua University of Education, Department of Geography, Changhua,
Taiwan ROC
[4]National Taichung University of Education, Department of Science Education and
Application, Taichung, Taiwan ROC
[#]Equal Contribution
[*]Corresponding author: Biqing Liang
(liangglobalcarbon@gmail.com;liangbq@mail.ncku.edu.tw)
**Abstract**
An approach for nano scale 3-D tomography of organic carbon (OC) and
associated mineral nano particles was developed to illustrate their spatial distribution and
boundary interplay, using synchrotron-based transmission X-ray microscopy (TXM). The
proposed 3-D tomography technique was first applied to in-situ observation of a lab-made
consortium of black carbon (BC) and nano mineral ($TiO_2$, 15 nm), and its performance
was evaluated under dual-scan absorption contrast and phase contrast modes. Then this
novel tool was successfully applied to a natural OC-mineral consortium from high
mountain soil at a spatial resolution down to 60 nm, showing the fine structure and
boundary of OC, distribution of abundant minerals at nano size, and in-situ 3-D organo-
mineral association. The stabilization of aged natural OC was found attributed to the



physical protection of Fe-containing minerals (Fe oxyhydroxides including ferrihydrite,
goethite, and lepidocrocite) of nano size, and the strong organo-mineral complexation.
The sorption of OC (and cation) to Fe oxyhydroxides through organo-mineral multiple
complex bonds such as 'ligand exchange' could occupy and consume their respective
reactive surface sites, tune down their activity and enhance their respective stabilization.
The ubiquitousness and abundance of mineral nano particles, and their high
heterogeneity in natural environment could have been seriously underestimated by
traditional study approach. Our in-situ description of organo-mineral interplay at nano
scale provides direct evidence to substantiate the importance of mineral physical
protection for OC long term stabilization. Mineral physical protection for OC stabilization
may be more important than previous understanding. This high resolution 3-D
tomography tool is promising for new insight on the interior 3-D structure of micro-
aggregates, in-situ organo-mineral interplay, and the fate of mineral nano particles
including heavy metals in natural environment.
**Introduction**
Mineral association with organic carbon (OC) may be an important stabilization
mechanism for carbon long-term sequestration, yet little is known about their in-situ
interplay and extent of association on aggregation level either chemically or physically
(Baldock and Skjemstad, 2000; Cusack et al., 2012; Mikutta et al., 2006; Torn et al., 1997;
Vogel et al., 2014). Traditional fractionation methods based on size and external force for
dissecting the association strength between OC and minerals in soils are limited to bulk



sample. High resolution information and in-situ knowledge is required for interpretation of
fractionation results and modeling (Kaiser et al., 2002; Kleber et al., 2007; Sollins et al.,
2009). Nano scale two-dimensional isotopic mapping discovered that only a limited
amount of the clay-sized surfaces contributed to OC sequestration (Vogel et al., 2014).
Understanding OC interplay with minerals in the fine fraction warrants study in a three
dimensional way (Kinyangi et al., 2006; Lehmann et al., 2007; Lehmann et al., 2008;
Solomon et al., 2012). Detailed in-situ association information between OC and minerals
may lead to breakthrough on mineral physical protection mechanism for OC long term
stabilization. To overcome the limitations of commonly used electron microscopic
methods (such as only on the surface layer, and undesirable artifacts due to
pretreatments), non-destructive high-resolution X-ray 3-D tomographic technique will be
used for exploring the fine structure of OC and boundary interplay with mineral nano
particles.
High resolution Synchrotron-based TXM has been demonstrated as a powerful
tool for understanding the internal 3-D structure of particles down to nano meter scale,
due to its large penetration depth and superior spatial resolution (Kuo et al., 2011; Wang
et al., 2015). This technique was successfully applied to reveal the discrete three
dimensional micro-aggregation structure of clay (kaolinite) in natural aqueous
environment, and generated remarkable tomography that revealed precise inter-particle
structure (Zbik et al., 2008) as well. Clay particles with diameter below 500 nm were
clearly visible and their pseudohexagonal symmetry was recognized in details in a three
dimensional way.



The synchrotron-based TXM at the beamline BL01B1 of Taiwan Light Source
(TLS), which has been used in this study, provides two-dimensional imaging and three-
dimensional tomography at a spatial resolution of 30/60-nm with tunable energy (8-
11keV). It provides unprecedented opportunity for studying OC boundary interplays with
mineral particles at nano meter scale. Two image acquisition modes, absorption contrast
and phase contrast, can be used alternatively for recognizing OC and nano minerals.
Conventionally, X-ray images are taken in the absorption contrast mode, and the resulting
image contrast only depends on the difference of X-ray attenuation coefficient between
materials. This mode is especially useful for materials consisted of high atomic number
compositions. However, in organic materials, the difference of X-ray attenuation
coefficients between specimen and air is too small to distinguish each other. For this
reason, the structure of organic materials is often hard to be recognized due to low
contrast in absorption contrast images. Alternatively, phase contrast technique transfers
optical path length differences (optical phase) inside specimens into intensity contrast,
can be used for imaging low atomic number materials, which are poor to absorb X-rays.
It provides a unique opportunity to observe fine structures of organic specimens such as
OC. Little study has been done on OC and mineral nano particles using high-resolution
3-D X-ray tomography, though non-synchrotron-based 3-D X-ray microscopy was used
to observe occluded carbon in phytolith structure and kerogen in micrometer scale
(Alexandre et al., 2015; Bousige et al., 2016). We aim to develop a new dual-scan method
using phase contrast and absorption contrast modes of the TXM alternatively for the
observation of OC and mineral consortiums inside lab-made and natural samples in nano
meter scale. Lab-made OC in forms of black carbon (BC) will be examined in the artificial



consortium with added nano mineral (TiO$_2$) particles using synchrotron-based TXM for
the first time.
Black C/biochar has received increasing research interest globally due to its
importance in global carbon cycling, soil fertility improvement and environmental pollutant
remediation (Bond et al., 2013; Jeffery et al., 2015; Kuhlbusch, 1998; Lehmann et al.,
2007; Liang et al., 2006; Liang et al., 2008; Schmidt, 2004). On top of method
development for 3-D tomography at nano meter scale, this study provides in-situ
evidences on the minerals physical protection on natural OC, and to explore the C
stabilization mechanism in natural soil.
**Methodology**
**Sample preparation and background**
Black C was made in lab using leguminous plant (*Sesbania roxburghii*) of 80 days'
harvest, which was first oven-dried (65 ℃) and charred inside a muffle furnace at 300 ℃
in loosely sealed stainless containers (Chen et al., 2014b). This consortium of low
temperature BC and mineral nano particles was constructed by dry deposition of
commercial TiO$_2$ (15 nm) on lab-made BC (3 mm chunk), and then embedded in Gatan
G-1 epoxy. The block were cross sectioned to a thickness of 100 to 200 μm using a
microtome (Leica Reichert Ultracut E ultra-microtome) and subsequently hand-polished
to a thickness of 30 to 50 μm. Each section was transferred onto Kapton tape and
mounted on a stainless steel sample holder for TXM observation. Before TXM analysis,



gold nano particles (50-150 nm or 400-500 nm in diameter) were deployed on the section
surface for image registration before 3-D tomography reconstruction.
Thin section of natural OC and mineral consortium (NH) was prepared using
micron to millimeter size particulate sample from high mountain soil. Particulate organic
matter of mm-size with minerals embedded inside was taken from the lower dark layer at
depth of 72-93 cm in a Typic Humicryepts soil profile, located in Mt. Nanhua, Nantou
County, Taiwan (24°03'00", 121°17'02"). On top of this dark layer, iron stain was observed
within the depth of 63-72 cm in the profile. The soil has developed on top of sandstone
and slate, with some features of inceptisol and spodosol. The sampling elevation is 3092
m, the annual temperature is 7.57 ℃, and the yearly rainfall is 2203.1 mm. The major
vegetation is arrow bamboo (*Yushania nittakayamensis*), with sporadic Hemlock (*Tsuga*
*chinensis*), fir (*Abies kawakamii*), and spruce (*Picea morrisonicola*).
The sequestration environment represents weak leaching and inactive chemical
weathering conditions. The age of soil organic C has been estimated to 3500 years B.P.
**Working conditions of TXM**
A superconducting wavelength shifter source provides a photon flux of $4\times10^{11}$
photons s$^{-1}$ (0.1% bw)$^{-1}$ in the energy range of 5-20 KeV at the BL01B1 beamline. A
double crystal monochromator exploiting a pair of Ge (111) crystals selects X-rays within
the energy range of 8-11 KeV. The specimen is imaged using a Fresnel zone plate, which
is used as an objective lens for an image magnification of 44× by the first order diffraction





mode. Conjugated with a 20× downstream optical magnification, the TXM provides a total
magnification of 880× with a field of view of 15×15 μm². By acquiring a series of 2D
images with the sample rotated 1º stepwise, 3-D tomography datasets is later
reconstructed based on 151 sequential image frames that are captured with azimuth
angle rotating from -75º to +75º.
**Image acquisition for 3-D tomography**

8         Under the most frequently used absorption contrast mode, 2-D images are

recorded based on the projection of the different X-ray absorption coefficient integration
along the optical pathway through samples on a detector. The absorption mode is useful
for materials of high absorption coefficient, such as minerals or high atomic number
materials, but it is poor for the observation of low atomic number materials, such as
organic or polymer materials. In order to recognize the OC structure more accurately, 2-
D/3-D images for the same sample region are recorded using absorption contrast and
phase contrast modes, respectively.

16        In the phase contrast mode, the gold-made phase ring positioned at the back-focal

plane of the zone plate is used to retard or advance the phase of the non-diffractive light
by π/2, generating (Zernike's) phase contrast images recording at the detector. The light
diffracted by specimen is interfered with the retarded non-diffractive light, generating
phase contrast image. The intensity difference in a phase contrast image shows the



combination of optical phase difference and absorption difference through specimens. This ability is especially important for the observation of OC which has a low X-ray absorption coefficient.

**3-D reconstruction and analysis**

Three dimensional tomography reconstruction is performed using homemade software, which is coded based on iterative image registration (Faproma) (Wang et al. 2017) and filtered back projection (FBP) reconstruction algorithms. Firstly, a serial of single TXM image captured from -75⁰ to +75⁰ at rotational increments of 1° is loaded to do image registration automatically using Faproma algorithm. Finally, the reconstruction is processed using FBP algorithm. The reconstructed dataset is exported in cross-sections, and later used for 3-D visualization using *Amira*. The intensity contrast of reconstructed datasets is inversed for better visualization; Compositions with higher absorption coefficients are shown in higher intensity and with low absorption coefficients are shown in lower intensity. The exported cross-section of 3-D tomography (reconstructed datasets) shows the real distribution details and boundary interplay of OC and mineral particles. The final 3-D tomographic structures for visualization and illustration are generated using *Amira* 3-D software for image post-process and computation (Fig. S1).

**Elemental mapping by SEM-EDS**



For correlated spatial distribution of selected elements (C, O, Fe, Al) in natural OC
particles from high mountain soil, a low-vacuum scanning electron microscope (JEOL W-
LVSEM, JSM-6360LV) equipped with an energy dispersive X-ray spectrometer (Oxford
EDS) and a cathodoluminescence (CL) image detector (Gatan mini-CL) was used for
elemental mapping, at an accelerating voltage of 15 KeV.

**7   X-ray Diffraction for Mineralogy**

To analyze the forms of minerals associated with natural OC, particulate OC (with
minerals on surface and embedded inside) was grounded and injected into capillary tubes
(Special Glass 10, Hampton Research, CA) for synchrotron high resolution X-ray
diffraction analysis at 09A beamline at Taiwan Photon Source (TPS), which is equipped
with a set of high-resolution monochromator (HRM). The wavelength is 0.8266 Å at the
energy of 15 KeV. The XRD spectra were recorded under room temperature for 240s
accumulation time and specific X-ray diffraction peaks and patterns were assigned ICDD
using PDF-2/4 program.

**17   Carbon functionality and interfacial mineral forms using SR-FTIR**

For FTIR analysis, mineral-bearing OC (NH) particles were grounded, dried (60 ℃
overnight), and mixed with potassium bromide (KBr) at a ratio of 1:100, and molded into
disks using a hydraulic press. During the pressing process, a vacuum pump was used for
evacuating   air   and   water.   The   samples   were   measured   using   Infrared



Microspectroscopy (IMS) at the BL14A1 beamline of the National Synchrotron Radiation
Research Center (NSRRC), Taiwan. The FTIR spectra were collected up to 1024 scans
in the mid-infrared range of 4000-400 $cm^{-1}$ with a spectral resolution of 4 $cm^{-1}$, using a
FTIR spectrometer (Nicolet 6700, Thermo Fisher Scientific, Madison, WI, USA) with a
self-equipped light source. The automatic atmospheric suppression function in OMNIC
(OMNIC 9.2, 2012; Thermo Fisher Scientific Inc., Waltham, MA, USA) for bulk sample
analysis was activated for data analysis, to eliminate the rovibration absorptions of $CO_2$
and water vapor in ambient air.
**Results and Discussions**
**Distinguish the fine structure of BC and boundary interplay with mineral nano**
**particles**
High resolution 2-D X-ray photographs were captured for the identical regions in
lab-made BC and nano mineral consortium using dual-scan absorption contrast and
phase contrast modes (Fig. 1, a, e). The cross-section views exported from the
reconstructed 3-D datasets reveal subtle details of BC and mineral nano particles, and
clearly outline the fine boundary of BC and the distribution of $TiO_2$ nano particles (Fig. 1).
The shape, size, and distribution of mineral nano particles can be identified accurately
using absorption contrast mode due to their high X-ray absorptivity (Fig.1, b, c and d). In
comparison, the BC structure and contour of its boundary can be revealed much more
clearly using phase contrast mode (Fig.1, f, g and h). However, the bright halo artifacts in
phase-contrast image enhance the intensity of margin texture for nano minerals, and may



lead to overestimation of their volume (Fig. 1, e, f, g and h). Use of dual-scan mode allows
cross-checking of details and validation.

3       Cross-section views of the reconstructed 3-D tomography share consistent and

comparable features of BC and nano minerals in multi-angles (Fig. 2). According to the
display of different slicing planes (XY, XZ, YZ), it can be recognized that $TiO_2$ nano
particles deposit inside BC only sporadically contact with BC boundary (Fig. 2, b, e, c,
and f) due to the treatment of dry deposition. The nano scale gap between BC and nano
minerals has been clearly observed in absorption and phase-contrast images (Fig. 2, b,
e, c, and f). It is feasible to calculate the interplay surface and mineral volume
quantitatively by examining each cross-section views in a selected region. Our approach
is a success in thorough exploration of OC and minerals 3-D distribution, and verification
of their real in-situ spatial correlation under nano scale resolution.
**3-D tomography for illustrating in-situ distribution of BC and mineral nano particles**

15       3-D tomography for visualization has been computed and generated to illustrate

the spatial correlation between BC and minerals based on post-process of reconstructed
3-D datasets. Unprecedented details of 3-D in-situ distribution of BC and mineral nano
particles are revealed in computed 3-D tomography (Fig. 3; Fig. SMOV1, 2). Results from
absorption mode and phase contrast mode are consistent and comparable. The fine
boundary feature of BC is contoured to a more completeness in the phase contrast mode.
The OC was rendered by transparent mode and high absorptivity materials (such as



minerals and gold particles) were rendered by solid mode with various colors. All
renderings are combined to visualize their interaction. The illustration of 3-D computed
tomography allows randomly tilted and set angles for image and animated video exports,
thus any region of interest inside a specimen may be explored thoroughly.

5        The lab-made consortium was successfully tested by this dual-scan methodology

using both absorption contrast and phase-contrast acquision modes (Figs. 1, 2, 3). Low
temperature BC, which is more similar to natural OC (especially recalcitrant OC) than that
made at high temperature, was especially made to test its applicability under absorption
contrast mode. Results show that the fine structure and boundary of low temperature BC
can be clearly observed under absorption contrast mode. Thus for environmental OC
samples, the use of absorption contrast mode is probably sufficient for capturing organo-
mineral features.

13       Different from field samples, the minerals observed within the lab-made consortium

often distribute in clusters and are only sparsely in association with BC surface. The
preservation of plant-like structures in BC could play a role for carbon stabilization in
natural environment, as their porosity and reactive surface provide large areas and sites
for mineral coating, which may contribute to their long residence and physical endurance
(Eusterhues et al., 2008; Rasmussen et al., 2005; Rawal et al., 2016).
**Interplay of OC and minerals and C stabilization in high mountain soil**



Nano scale 3-D tomography provides new insight for the mineral physical
protection mechanism of OC in soil. Natural OC exhibited strong organo-mineral
association on its surface at nano scale in high mountain soil (Fig. 4; Fig. SMOV3).
Abundant short-range order minerals in forms of subhedral particle or anhedral nano-
aggregate have direct contact with the boundary of OC and develop coating on the
tracheid surface (Fig. 4 b and c) (Mikutta et al., 2006). Mineral aggregation by poorly
crystalline nano particles renders natural sub-micron porosity, which may contribute to
elevated sorption capacity in soil (Rawal et al., 2016). The densely-packed mineral texture
suggested significant physical protection on OC surface (Kaiser and Guggenberger,
2007). The sorbed minerals not only can form physical protection, but also could shield
OC from chemical weathering (Mikutta et al., 2006).
The nature of associated minerals was confirmed to be mainly Fe oxyhydroxides,
specifically ferrihydrite (ICDD 01-073-8408), goethite (ICDD 01-073-6522), lepidocrocite
(ICDD 00-044-1415), and quartz (ICDD 00-033-1161) (Fig. 5; Table S1), analyzed using
high resolution synchrotron-based XRD. Quartz may be at most a minor component on
OC surface, considering their chemistry and particle size. Yet siliceous mineral surfaces
may become coated with a veneer of hydrous Al- and Fe- oxides, which could confer net
positive charge and promote their reactivity in tropical environments (Chen et al., 2014a;
Sposito, 1989).
Considering their high surface area and reactivity, the abundant nano scale Fe
oxyhydroxides could play a significant role for OC long-term stabilization through
chemical bonding and physical shielding (Eusterhues et al., 2005; Kaiser et al., 2002;

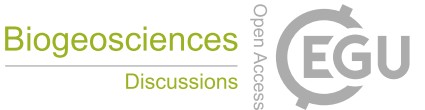

Kiem and Kogel-Knabner, 2002; Mikutta et al., 2006), as well as cation sorption in soil,
and contribute to longevity of OC in high mountain soil. According to elemental mapping
results, aluminosilicates may also be present, however, their portion and crystalline level
should be low due to their minimal signal in the XRD spectra (Figs. 5, 6). The FTIR
analyses reveal the chemistry of organo-mineral association (Fig. 7; Table S2). The aged
OC is highly aromatic when at the same time highly reactive, as broad bands centered at
1596 $cm^{-1}$ and 1706 $cm^{-1}$ for aromatic C=C stretching and $\nu$ C=O in carboxyl are
observed , pointing to likely origin of pyrogenic C (Özçimen and Ersoy-Meriçboyu, 2010;
Sharma et al., 2004). Both aromatic and carboxyl C functional groups normally have high
affinity with Fe (III) (Mikutta et al., 2007; Zhao et al., 2016). The broad bands point to
possible significant degree of association between OC and minerals (Chen et al., 2016;
Gu et al., 1994; Kaiser and Guggenberger, 2007). The sorption of OC to Fe oxyhydroxides
through organo-mineral multiple complex bonds such as 'ligand exchange' could occupy
and consume the reactive surface sites on OC and Fe oxyhydroxides, tune down their
activity and enhance their respective stabilization (Chorover and Amistadi, 2001; Cornell
and Schwertmann, 2006; Hall et al., 2016; Kaiser and Guggenberger, 2007; Mikutta et
al., 2007). The discovery of short-range-order mineral ferrihydrite in air-dried OC particles
and later ground samples indirectly validates its stabilization due to organo-mineral
interplay. As a metastable mineral, ferrihydrite is hard to estimate accurately in dry soil
samples due to its transient nature and the limitation of traditional extraction and
spectroscopic methods (Cornell and Schwertmann, 2006). The specific mineral forms in
direct contact with OC on surface at nano scale warrants future study (Fig. SMOV3). In-



situ mineral mapping of different Fe oxyhydroxide on OC surface will provide mechanistic
evidence on OC stabilization. Mineral physical protection on OC may represent the end
stage of carbon stabilization, especially in a weak leaching and weathering environment.
Our in-situ description of organo-mineral interplay at nano scale provides direct
evidence on the important role of mineral physical protection for OC long term stabilization.
High amounts of ferrihydrite and other Fe oxyhydroxides were also found associated with
lignin-like OC in soil under an aquic moisture regime (Eusterhues et al., 2011). The
abundance of mineral nano particles, and their high heterogeneity and short-range-order
nature could be common in humid environment, however, they could have been seriously
underestimated by traditional analysis methods (Mikutta et al., 2005). Mineral physical
protection for OC stabilization may be more important than previous understanding.
In summary, a high resolution 3-D tomography tool is required for exploring the in-
situ interplay of OC and nano minerals in natural environment. Nano scale 3-D
tomography provides direct evidence and new insight for the mineral physical protection
mechanism of OC in soil. This high resolution 3-D tomography approach is a promising
technique for probing the multi interfacial features between OC and minerals in lab and
field samples, and may provide new perspective on the fate of nano particles including
heavy metals in natural environments.
**Figure Captions**



**Figure 1.** The 2-D X-ray images for the same region of BC and mineral nano particle
consortium obtained using absorption contrast mode **(a)** and phase contrast mode **(e)**.
Cross-section views of the reconstructed 3-D tomography under each mode at different
depths relative to the position of gold nano particle along Z-axis as a reference. **(b)** and
(**f**) are sections extracted at the position of the gold particle. **(c)** and **(g)** are sections
extracted at 800 nm above the gold particle. **(d)** and **(h)** are sections extracted at 800 nm
below the gold particle. The scale bar is 5 µm.
**Figure 2.** Three-directional orthogonal sections of lab-made BC and mineral nano particle
consortium. The upper row sections are extracted from absorption contrast tomography
**(a, b, c)**, and the lower row sections are extracted from phase contrast tomography **(d, e,**
**f),** specifically (**a**) and (**d**) are for XY plane, (**b**) and (**e**) are for YZ plane, and (**c**) and (**f**)
are for XZ plane. The scale bar is 5 µm.
**Figure 3.** 3-D tomography illustration of lab-made BC and mineral nano particle
consortium observed at -45° (**a**, **d**), 0° (**b**, **e**), and +45° (**c**, **f**) azimuthal viewing angles
under absorption contrast (**a**, **b**, **c**) and phase contrast mode (**d**, **e**, **f**). The scale bar is 5
µm.
**Figure 4**. Three-directional orthogonal sections of high mountain mineral-bearing OC
from absorption contrast tomography (**a** for XY plane, **b** for XZ plane, and **c** for YZ plane). .
The scale bar is 5 µm. Minerals mainly present two types of textures, subhedral particles
and anhedral nano-aggregates. The lower row images highlight the free surface of
specimen (red line in **d**), the boundary of OC (green dotted-line in **e**), and the subhedral
mineral particles (pink arrow in **e** and **f**)



**Figure 5**. The X-ray diffraction pattern of minerals within OC particles from high mountain soil. Highly reactive Fe oxyhydroxides are identified and denoted with lines of different colors: ferrihydrite (ICDD 01-073-8408, orange), goethite (ICDD 01-073-6522, blue), and lepidocrocite (ICDD 00-044-1415, green). Q stands for Quartz (ICDD 00-033-1161). Details are included in Table S1.

**Figure 6.** Elementary mapping by SEM-EDS for mineral-bearing OC from high mountain soil. Left: SEM backscattering image (The bright spots inside are gold nano particles for coating). Right: Elemental mapping of C, O, Fe and Al. Scale bars are 20 µm.

**Figure 7**. The FTIR spectra for the chemistry of organo-mineral association. The aged OC is highly aromatic (1596 and 1386 cm$^{-1}$), and highly reactive with obvious carboxyl functional group (1706 cm$^{-1}$). The broad bands point to possible significant degree of association between OC and minerals. Some minor bands near 1274, 1062, 1024, and 989 cm$^{-1}$ indicate the lignin-derived nature of OC. Those bands near 476, 534, 798, 910 and 1025 cm$^{-1}$ have similar characteristics of soil inorganic/mineral matrix. More details are included in Table S2.

**Acknowledgement**

We thank Dr. Chung-Ho Wang for kind support, the technical support from Ms. Hsueh-Chi Wang (TXM, TLS-BL01B01); Dr. Yao-Chang Lee and Ms. Pei-Yu Huang (FTIR, TLS-BL14A1); and Dr. Hwo-Shuenn Sheu and Dr. Yu-Chun Chuang (XRD, TPS-09A1) at the



end-stations of NSRRC (Taiwan), the help for SEM-EDS analysis from Dr. Yoshiyuki
Iizuka (Academia Sinica), the SC specimen from Dr. Chih-Hsin Cheng (National Taiwan
University), the $TiO_2$ nano particles from Dr. Yen-Hua Chen (NCKU, the department of
Earth Sciences), and Dr. Chia-Chuan Liu, the former and current members of the NCKU
Global Change Geobiology Carbon Laboratory for help and support.

**Funding Sources**

BQ Liang and CC Wang acknowledged the funding support from Taiwan Ministry of
Science and Technology (MOST 102-2116-M-006-018-MY2, MOST 105-2116-M-006-
010-, and MOST 105-2112-M-213-001).

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

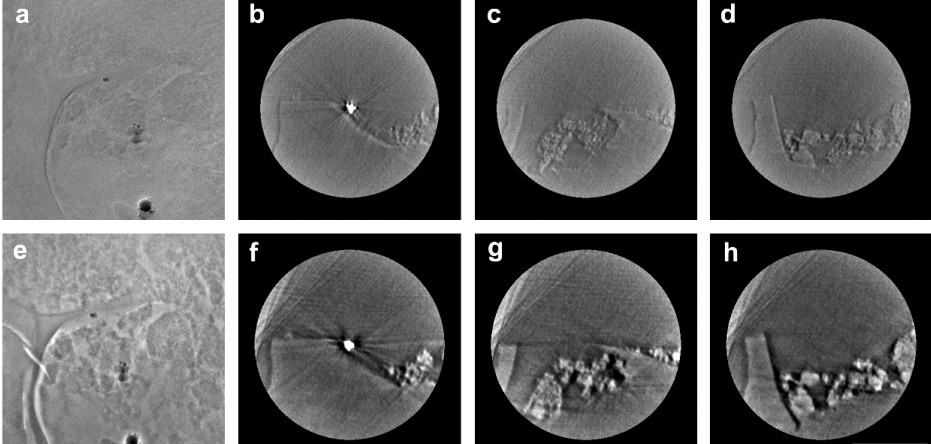

18  **Figure 1.**



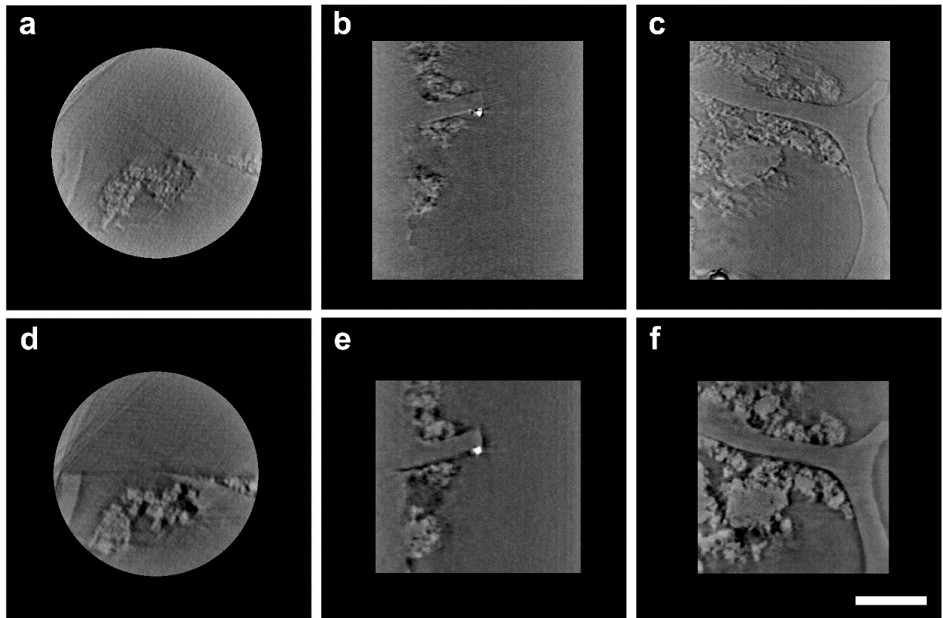

**Figure 2.**




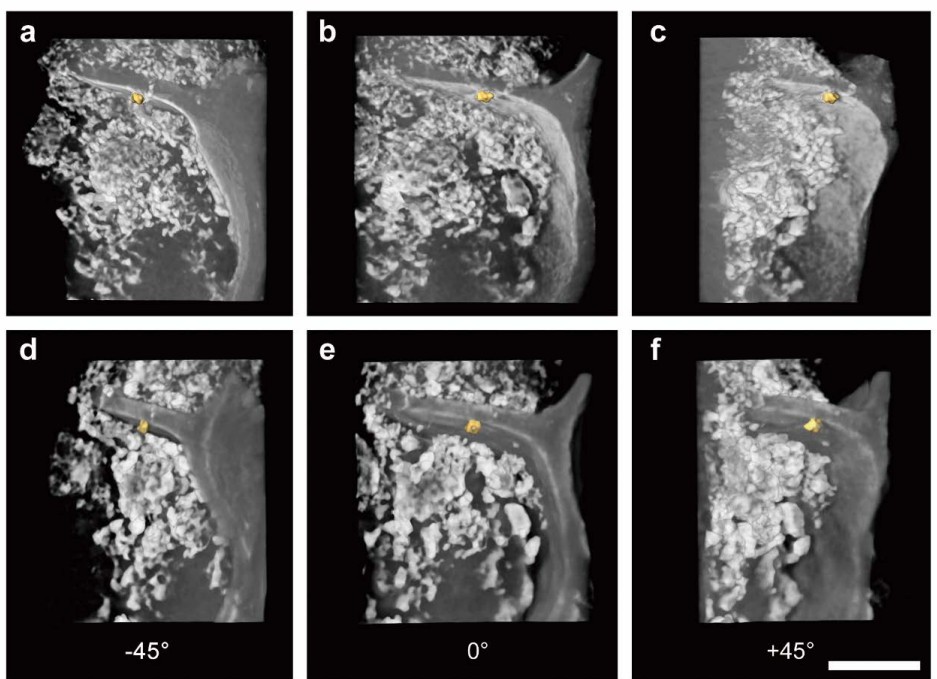

2   **Figure 3.**





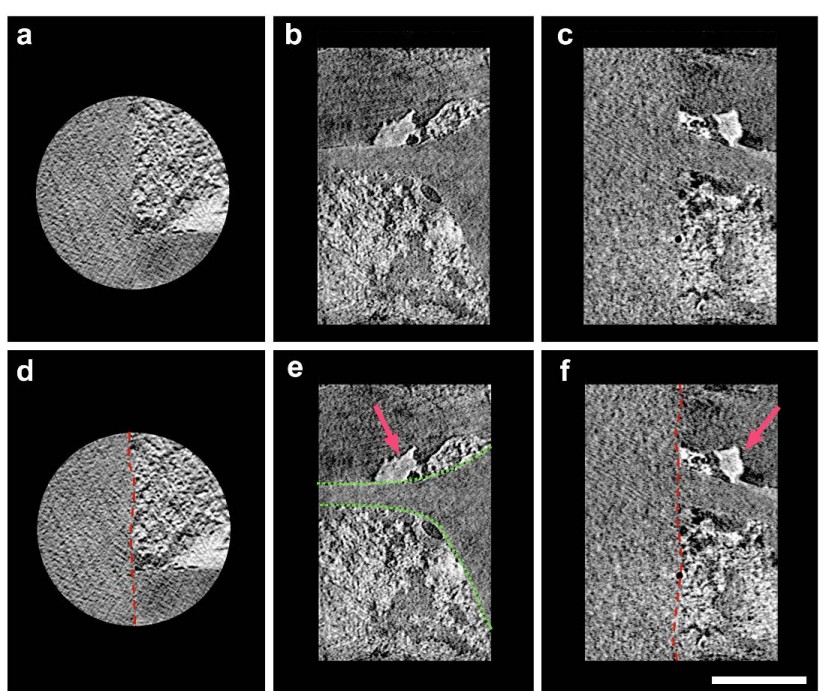

3   **Figure 4**.



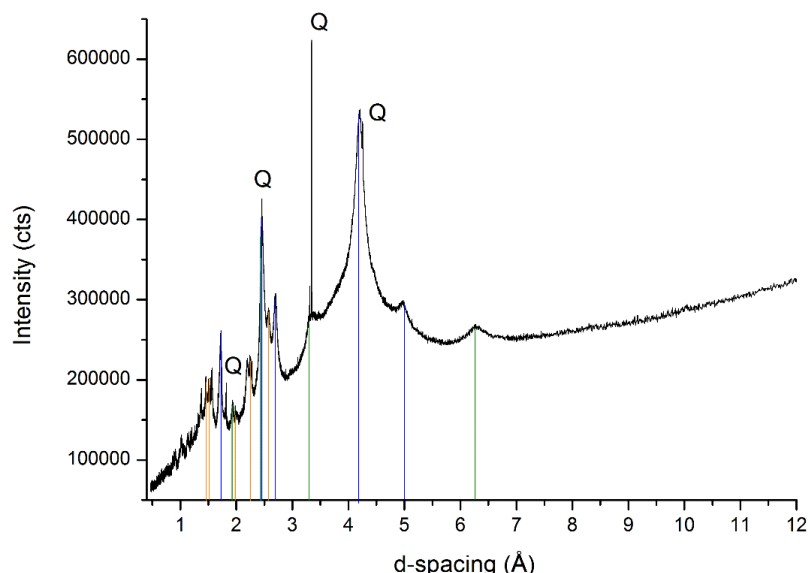

2    **Figure 5**.

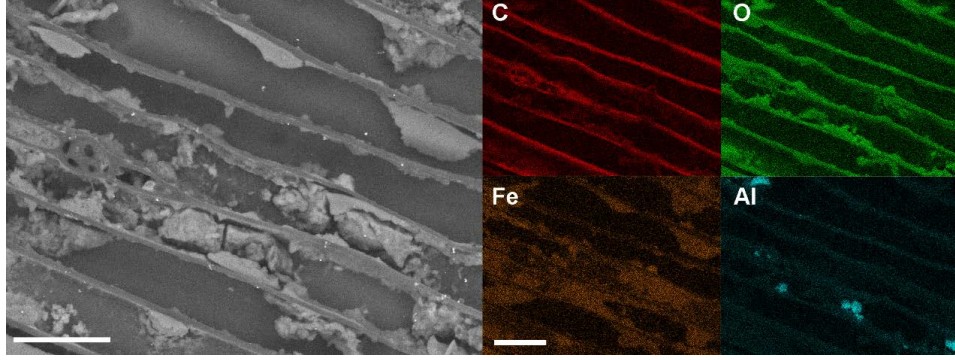

6    **Figure 6**.





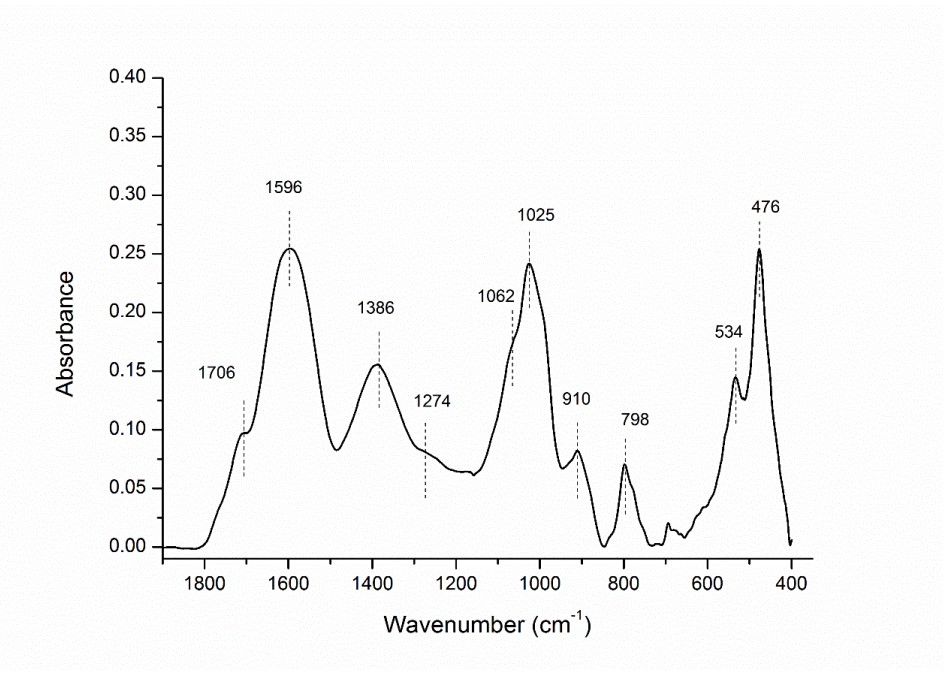

2      **Figure 7**.

