# Peer review of "Mineral physical protection and carbon stabilization in-situ evidence"

_Biogeosciences, 2017_

## Referee Comment (RC1) · Anonymous Referee #1 · 17 Jan 2018

In this paper, Weng el al study the 3D distribution of organic carbon and mineral particle using synchrotron-based transmission X-ray microscopy. The approach was applied to a lab-made mixture of black carbon and nano mineral (TiO2) and to a natural soils rich Fe oxyhydroxides. By observing criss-cross of mineral particles and organic matter they conclude on the importance of mineral physical protection (for example through ligand exchanges) for long-term persistence of OC. They also conclude that mineral physical protection for OC stabilization may be more important than previously thought. I have the feeling that this study uses a Jack Hammer to open nuts. We already know that minerals play key role in the persistence of organic matter in soils. The specific role of reactive minerals such as Al and Fe oxyhydroxides, allophane is known since

several decades. Therefore, I do not see what progress in our knowledge is made here. Moreover, the observation of criss-cross of mineral particles and organic matter in one soil layer does not inform on the role of minerals on OM persistence and does not permit any generalization. We do not know whether the OM is really retains by minerals, with what forces and what consequences for its fate in soils. From my modest knowledge, I think that what we need now is to determine whether the capacity of minerals to fix carbon is limited; If this capacity is limited, how much carbon can still be stored in soils (by considering the whole soil profile). Understanding how some organic compounds free of minerals can persist over centuries (An example among many others: Derenne et al. 1991) could be another original and exciting research avenue.

---

## Referee Comment (RC2) · Anonymous Referee #2 · 7 Feb 2018

This study aimed to develop the 3-D tomography of organic carbon-mineral consortium using the synchrotron based transmission X-ray microscopy. Both lab-made and natural black carbon were tested. Results of in-situ 3-D tomography directly indicate the association between OC and minerals in both samples. For the natural black carbon, other spectroscopic results including XRD and FTIR also demonstrate the abundance of Fe (hydr)oxides and their significance in the physical stabilization of OC. Although the association between Fe minerals and OC has been well known for decades, this study showed the direct and clear evidence for such mechanism. The contribution of this study lies in the fact that authors developed the methodology to obtain the 3-D images using the TXM. In terms of the C sequestration, the degree of C stabilization

is related to the structural development of Fe/C assemblages. Previous studies could only gauge the adsorption vs. coprecipitation between OC and Fe minerals via the indirect spectroscopic evidences. The 3-D technique developed here provides a new insight to determine the structure of such micro-aggregates. In this case, I would suggest authors to add more information regarding the differentiation of adsorption vs. coprecipitation between OC and Fe minerals using the 3-D tomography results to shed light on its significance toward the C stabilization.

---

## Author Comment (AC1) · 14 Feb 2018

Reply to Reviewer#1: We have carefully prepared our response, it is lengthy, and we appreciate the patience of associate editor and reviewer #1. Revised manuscript is also attached (shown as a supplement file).

Abbreviation: organic carbon (OC), short-range-order (SRO), soil organic matter (SOM), soil organic carbon (SOC)

We use an old (3500 years) pyrogenic OC-mineral consortium to demo the in-situ OC-mineral interplay in a relatively simple mineralogy context. In-situ evidence reveals abundant mineral nano particles, in dense thin layers or nano-aggregates/clusters, instead of crystalline/micron/clay-size minerals on or near OC surfaces, the key working minerals for C stabilization are essentially SRO minerals and/or poorly crystalline sub-micron size clay minerals. Supported by spectroscopic results, the studied OC is not merely in criss-cross co-localization with reactive SRO minerals. There is a significant degree of binding between OC and minerals. We have other sets of unpublished 3-D tomography data for samples from various mineralogy background, and the minerals found associated with OC surface are coincidentally in line with nano particles/poorly crystalline minerals. 'Mineral physical protection for OC stabilization may be more important than previously thought' is not a solid conclusion, the statement is deleted.

Technology Progress for In-Situ Evidence:

This high resolution 3-D tomography tool we introduce has unprecedented resolution, is promising for new insight on in-situ organo-mineral association, the interior 3-D structure of microaggregates, and the fate of mineral nano particles including heavy metals in natural environment. Though some 3-D tomography research have been done on soil microstructure and porosity using X-ray micro-CT, the best achieved resolution only reaches tens of microns (Quin et al., 2014 (with biochar amendment, 70 microns); Kravchenko et al., 2015 (13 microns)), which exceeds the size of clay minerals (<2 microns), and not fit for capturing the microstructure of clay and the submicron assemblage of SRO minerals such as Fe oxyhydroxides. High resolution X-ray scanning and 3-D tomography is highly demanding in terms of technology, multidisciplinarity and big-data analysis (Lafond et al., 2015). Except for the adoption of high resolution X-ray objective, fidelity of 3D tomography relies on the accurate alignment of the 2D projections in correct three-dimensional positions. However, non-negligible mechanical imperfection of the rotational stages at nanometer level, or the thermal effects may significantly degrade the spatial resolution of reconstructed tomography. We've developed a markerless image auto-alignment algorithm for fast projection matching (Faproma, Wang et al., 2017) to surpass these difficulties, and accomplished accurate reconstruction of

3-D tomography down to nano meters.

Carbon Stabilization:

Soil organic C can be: (1) physically stabilized, or protected from decomposition, through microaggregation, or (2) intimate association with silt and clay particles in organo-mineral complexes, and (3) can be biochemically stabilized through the formation of recalcitrant SOC compounds (Six et al., 2002).Three main mechanisms of Soil organic matter stabilization are: (1) chemical stabilization, result of chemical or physiochemical binding between SOC and soil minerals (aka organo-mineral complexation), especially clay and silt in current opinions, (2) physical protection, which is predominantly at the microaggregate level, built on top of chemical organo-mineral complexation, and (3) biochemical stabilization, which is often experimentally defined, or equated to the nonhydrolyzable fraction (Stevenson, 1994; Christensen, 1996; Six et al., 2002). Experimentally to quantify C of different stabilized levels, a size/density fractionation approach with external force is routinely used for bulk soil, and three or more density fractions are divided: (a) free particulate OM, (b) occluded particulate OM, and (c) colloidal or clay associated OM, in the fine and dense fraction (Golchin et al., 1994, Sollins et al., 2009).

The stabilization of OC by the soil matrix is considered a function of the chemical nature of specific soil mineral fraction, the presence of multivalent cations, the presence of mineral surfaces capable of adsorbing organic materials, and the micro architecture of the soil matrix (Edwards and Bremner, 1967). The degree and amount of protection attributed by each mechanism depends on the chemical and physical properties of mineral matrix, the morphology, chemical nature and structureof organic matter (Baldock and Skjemstad, 2000). However, each mineral's unique capacity to stabilize organic C is rarely recognized in bulk sample analysis, or within size/density fractionation. Thus, quantifying the protective capacity of a soil requires a careful consideration of all mechanisms of protection and the implications of experimental procedures.

It is found that accumulation and subsequent loss of organic C were largely driven by changes in the millennial scale cycling of mineral-stabilized carbon (Torn et al., 1997, Nature).The protection of SOC by silt and clay particles in soils is well established for a positive relationship between old/stabilized organic C and the content of silt and/or clay in fine and dense fraction (Sorensen, 1972; Feller and Beare, 1997; Hassink, 1997; Trumbore, 1997). In addition to the clay content, clay type (2:1 vs 1:1 clay vs allophanic /short-range-order (SRO) minerals) has strong influence on the stabilization of organic C (Torn et al., 1997). Yet little is known about how spatial and temporal variation in soil mineralogy (especially type, phase, reactivity) controls C long term stabilization in terms of quantity and turnover (Oades, 1994, Torn et al., 1997). Quantification of interactive relationships is imperative, measurement and information using 2-D and 3-D quantitative tool will have great benefit in understanding soil C and aggregate dynamics (Six et al., 2004).

Very limited information is available on the in-situ distribution of clay-size or reactive minerals in soil, and subsequent organo-mineral micro assemblage in soils. For example, a substantial parts of mineral surfaces are considered likely not covered by organic matter (Ransom et al. 1998; Arnarson and Keil 2001; Mayer and Xing 2001; Kahle et al. 2002). Recently, nano scale two-dimensional mapping using Nano SIMS generated direct evidence, revealing that less than 19 percent of the clay-sized mineral surfaces co-localize with OC (Vogel et al., 2014, Nature Communication). Views of carbon sequestration in soils and the widely used carbon saturation estimates has been revolutionized to recognize likely only a limited proportion of the clay-sized surfaces contribute to OM sequestration in real soil environment. However, this research could underestimate the actual contact surface area between OC and mineral due to the limitation of sample treatment and two-dimensional approach. Exactly which reactive mineral in the clay-sized fraction interact with OC remains unknown. The cornerstone research done by Torn et al. (1997) revealed a positive relationship between non-crystalline minerals and organic carbon in soils through the climate gradient. In couple with extraction using ammonium oxalate and/or sodium dithionite-citrate-bicarbonate (DCB), SRO minerals, which may only be a small portion in the fine and dense fraction, are found of great importance for C stabilization, and further discovered preferentially associated with aromatic/lignin-like C (Kaiser et al., 2002; Eusterhues et al., 2005; Kleber et al., 2005; Mikutta et al., 2005, 2006, 2007; Rasmussen et al., 2005; Schneider et al., 2010; Cusack et al., 2012). The SRO minerals are known of high reactivity due to small size, large surface area and charges, their encounter with reactive OC surface may develop strong organo-mineral complexation (Eusterhues et al., 2008, 2011). Yet, the actual key mineral player (micron size clay vs nano scale SRO minerals) for interaction with OC, and the spatial and temporal variation of such minerals is not known in real soil environment (Vogel et al., 2014). On the other hand, a three-dimensional functional view of carbon turnover dynamics at the microscale has gained ground (Kleber et al., 2018), which consider a multitude of largely independent microreactors within soil. For example, anaerobic microsites are discovered of great importance for C stabilization (Keiluweit et al., 2017). New perspectives call for observational evidence on OC-mineral micro assemblage in nano scale details.

Using high resolution 3-D tomography and synchrotron-based spectroscopic approach, we unveil: i) There is a high heterogeneity within OC-mineral consortium, including many nano/micron size porosity and microsites. ii) Rare particulate OC surface is not coated by minerals. iii) The minerals on the OC-mineral interplay surface are essentially nano particles SRO minerals, or submicron/poorly crystalline minerals, which are highly reactive and carry large interactive surface for C absorption. iv) Very few micron size/clay-size, euhedral/crystalline minerals exist. v) The 3500-year-old pyrogenic OC is highly aromatic (1596 and 1386 cm-1), and also has highly reactive carboxyl functional group (1706 cm-1). vi) Broad FTIR bands points to a significant degree of organo-mineral association between OC and SRO minerals, which are not just criss-cross co-localization. vii) The OC surface is often coated by thin layers of SRO minerals, pointing to likely mineral absorption. viii) Within regions microns from particulate OC surface, abundant mineral nano particles develop nano-aggregates/clusters, pointing to likely co-precipitation of relatively free OC and SRO minerals.

Our results suggest:

i) The coated particulate OC surfaces by SRO minerals are stabilized. ii) Minerals physical protection contributes to long term persistence of OC (3500 year old) in the sub-tropical environment. iii) Substantial mineral surfaces are likely available for C stabilization. iv) Large amount of nano-aggregates/clusters likely interact/co-precipitate with relatively free OC, which is difficult to prove directly. v) Current electron microscopy/spectroscopy/fractionation approach focusing on clay-sized may lead to underestimate C stabilization attributed by mineral physical protection.

We propose:

i) High resolution 3-D tomography is a powerful, suitable tool for nailing down the correct target (key working mineral) for C stabilization. ii) Two-dimensional TEM may be applicable in revealing nano particle, amorphous/poorly crystalline minerals, on certain focal plane, but not for a whole view. iii) XRD is powerful for crystalline minerals but very poor in resolving SRO, cautions should be taken in mineralogy pattern interpretation and sample preparation. iv) More research should be done to prove: i) whether it is a general phenomenon that the minerals interact with OC are essentially nano particle SRO minerals/poorly crystalline submicron minerals, ii) whether the major mineral for C stabilization is nano scale SRO minerals instead of clay-sized minerals. v) Perspective on SOM dynamics may have to take account the role of SRO minerals into modelling parameters, instead of drawing conclusion mainly on clay type and content (Kleber et al., 2010).

Carbon Models and Carbon Saturation:

Most current models of SOM dynamics try to describe soil heterogeneity by defining several pools, typically three to five, experimentally applicable with supposedly different intrinsic decay rates. Meanwhile, first order kinetics are assumed for the decomposition of different conceptual pools (Parton, et al., 1994; Paustian, 1994; McGill, 1996), which infers equilibrium C stocks are linearly proportional to C inputs (Paustian et al.,

1997). These models predict that soil C stocks can theoretically be increased without limit, provided that C inputs increase without limit, leading to no assumptions of soil C saturation. Such models have been largely successful in representing SOM dynamics under some studied conditions and management practices, and native soils (e.g. Parton et al., 1987, 1994; Paustian et al., 1992; Powlson et al., 1996). Notably, the representation of the model pools (or quality spectrum) is primarily conceptual in nature (Christensen, 1996; Elliott et al., 1996; Six et al., 2002). The individual pools are generally only loosely associated with measurable quantities obtained with existing analytical methods.

The notion of carbon saturation is developed on the assumption that mineral physical protection could be limited by soil physiochemical characteristics such as silt and/or clay content, microaggregate, and surface area (Kemper and Koch, 1966; Hassink, 1997). Six et al. (2002) proposed a conceptual model which use the physicochemical characteristics inherent to soils to define the maximum protective capacity of different SOC pools, which may limit the increases in C sequestration with increasing organic residue inputs. This conceptual model includes four measurable pools: (1) a biochemically-protected C pool, (2) a silt and clay-protected C pool (less than 53 micron organomineral complexes) (3) a microaggregate-protected C pool (53 to 250 micron aggregates), and (4) an unprotected C pool. Each pool is supposed to has its own dynamics and stabilizing mechanism, which is in turn hypothesized to determine a level at which soil C becomes saturated. The silt and clay protected C pool, which is protected by association with the mineral particles, is defined as hydrolysable fraction and the biochemically-protected C pool is experimentally defined as nonhydrolyzable fraction. Biochemical stabilization is understood as the stabilization of SOM due to its own chemical composition (e.g. recalcitrant compounds such as lignin and polyphenols) and through chemical complexing processes (e.g. condensation reactions) in soil. The unprotected C pool is considered labile, an important nutrient source, which is very sensitive to management practices. Interestingly, the hierarchy for carbon stabilization based on protection capacity/level, from low to high has been ranked as: silt +

clay protected pool, microaggregate protected pool, biochemically protected, and non-protected pool on the top to cap the maximum carbon saturation. Six et al. (2002) pointed that there are major gaps in knowledge and proposed a research priority in terms of soil C saturation on mechanistic level, if it exists, especially for unprotected and biochemically protection pools.

As an outsider, we share a few points:

i) The heterogeneity of microaggregates is little understood and minimal quantitative data is available. ii) The unprotected pool actually involves some mechanism separation and dispersion, which is not truly unprotected. iii) Six et al. (2000) shared our thought that non-hydrolyzable fraction is not only biochemically stabilized but is also partially stabilized by association with clay and silt particles, and the silt and clay protected pool could also be partially stabilized by incorporation in microaggregates. iv) Our studied aged pyrogenic OC (3500 years) is not only biochemically stable, but also partially stabilized by association with submicron minerals.

Reviewer #1 suggested some interesting research on biochemically stabilized organic compounds free of minerals, which appears a vast expansion to very solid organic geochemistry. Many models for SOM dynamics indeed incorporate a huge 'inert' pool, which may include biochemically stabilized organic compounds free of minerals. Perspective on carbon stabilization may have to expand to a much longer geological time scale, as common C saturation models are dealing with a time scale of millennials. Derenne et al. (1991) carried out research on biochemically stabilized organic compounds free of minerals, especially on kerogen/fossil-like organic compounds, in terms of kerogen/fossil formation/biochemical transformation/condensation process. We suggest research on biochemically protected pool also expand into black carbon/pyrogenic organic carbon, which are biochemically stable due to their high aromaticity and have similar structure with kerogen and coal, can persist over millennials under natural exposure. Long term abiotic/biotic degradation, aerobic/anaerobic alternation can render development of reactive surface functional groups (Liang et al.,

2006), and lead to consequent mineral physical protection and C stabilization over millennials, even in the topics (Liang et al., 2008).

Please also note the supplement to this comment:
https://www.biogeosciences-discuss.net/bg-2017-509/bg-2017-509-AC1-supplement.pdf

**Supplement:**

[revised manuscript text omitted]

Nano scale 3-D tomography revealed a high heterogeneity within OC-mineral consortium and most particulate OC surface is coated by minerals.  Natural OC exhibited strong organo-mineral association on its surface at nano scale in high mountain soil (Fig. 4; Fig. SMOV3). Abundant short-range order minerals in forms of subhedral particle or anhedral nano-aggregate have direct association with the boundary of OC, and develop coating on the tracheid surface (Fig. 4 b and c) (Mikutta et al., 2006). Sheet-like mineral coating is observed on OC surface, which are dense and thin layers, likely originated from absorption. Another distinct texture is recognized as nano-aggregates/clusters of various shapes, possibly formed by OC-mineral co-precipitation. Mineral aggregation by poorly crystalline nano particles renders natural sub-micron porosity, which may contribute to elevated sorption capacity in soil (Rawal et al., 2016). The densely-packed mineral texture suggested significant physical protection on OC surface (Kaiser and Guggenberger, 2007). The sorbed minerals not only can form physical protection, but also could shield OC from chemical weathering (Mikutta et al., 2006). The key working minerals for OC-mineral interplay are essentially nano particles/submicron/SRO/poorly crystalline minerals.

The nature of associated minerals was confirmed to be mainly SRO Fe oxyhydroxides, specifically ferrihydrite (ICDD 01-073-8408), goethite (ICDD 01-073-6522), lepidocrocite (ICDD 00-044-1415), and quartz (ICDD 00-033-1161) (Fig. 5; Table S1), analyzed using high resolution synchrotron-based XRD. Quartz may be at most a minor component on OC surface, considering their chemistry and particle size. Yet siliceous mineral surfaces may become coated with a veneer of hydrous Al- and Feoxides, which could confer net positive charge and promote their reactivity in tropical environments (Chen et al., 2014a; Sposito, 1989).

Considering their high surface area and reactivity, the abundant nano scale Fe oxyhydroxides could play a significant role for OC long-term stabilization through chemical bonding and physical shielding (Eusterhues et al., 2005; Kaiser et al., 2002;

Kiem and Kogel-Knabner, 2002; Mikutta et al., 2006), as well as cation sorption in soil, and contribute to longevity of OC in high mountain soil. According to elemental mapping results, aluminosilicates may also be present, however, their portion and crystalline level should be low due to their minimal signal in the XRD spectra (Figs. 5, 6). The FTIR

analyses reveal the chemistry of organo-mineral association (Fig. 7; Table S2). The aged

OC is highly aromatic when at the same time highly reactive, as broad bands centered at

$cm^{-1}$ and 1706 $cm^{-1}$ for aromatic C=C stretching and $\nu$ C=O in carboxyl are observed , pointing to likely origin of pyrogenic C (Özçimen and Ersoy-Meriçboyu, 2010;

Sharma et al., 2004). Both aromatic and carboxyl C functional groups normally have high affinity with Fe (III) (Mikutta et al., 2007; Zhao et al., 2016). The broad bands point to possible significant degree of association between OC and minerals (Chen et al., 2016;

Gu et al., 1994; Kaiser and Guggenberger, 2007). The sorption of OC to Fe oxyhydroxides through organo-mineral multiple complex bonds such as 'ligand exchange' could occupy and consume the reactive surface sites on OC and Fe oxyhydroxides, tune down their activity and enhance their respective stabilization (Chorover and Amistadi, 2001; Cornell and Schwertmann, 2006; Hall et al., 2016; Kaiser and Guggenberger, 2007; Mikutta et al., 2007). The discovery of short-range-order mineral ferrihydrite in air-dried OC particles and later ground samples indirectly validates its stabilization due to organo-mineral interplay. As a SRO/metastable mineral, ferrihydrite is hard to estimate accurately in dry soil samples due to its transient nature and the limitation of traditional extraction and spectroscopic methods (Cornell and Schwertmann, 2006). The specific mineral forms phase in direct contact with OC on surface at nano scale warrants future study (Fig. SMOV3). In-situ mineral mapping of different SRO minerals/ Fe oxyhydroxide on OC surface will provide mechanistic evidence on OC stabilization. Mineral physical protection on OC may represent the end stage of carbon stabilization, especially in a weak leaching and weathering environment.

Our in-situ description of organo-mineral interplay at nano scale provides direct evidence on the important role of mineral physical protection for OC long term stabilization. High amounts of ferrihydrite and other Fe oxyhydroxides were also found associated with lignin-like OC in soil under an aquic moisture regime (Eusterhues et al., 2011). The abundance of mineral nano particles, and their high heterogeneity and short-range-order nature could be common in humid environment, however, they could have been seriously underestimated by traditional analysis methods, such as electron microscopy/X-ray diffraction/fractionation approaches, which focus on clay-size minerals (Mikutta et al., 2005). Mineral physical protection for OC stabilization may be more important than previous understanding. More research is proposed to explore: i) whether it is a general phenomenon that the minerals interact with OC are essentially nano particle/submicron/SRO/poorly crystalline minerals, ii) whether the major mineral for C stabilization is nano scale SRO instead of clay-size mineral in soils. Perspective on C stabilization and saturation should take account the role of SRO minerals into soil C dynamics modelling, besides clay type and content.

[revised manuscript text omitted]

---

## Author Comment (AC2) · 14 Feb 2018

Reply to Reviewer#2:

We thank Reviewer#2 for recognizing our work in methodology development and contribution to the understanding of C stabilization. In-situ evidence reveals abundant mineral nano particles, in dense thin layers or nano-aggregates/clusters, instead of crystalline/micron/clay-size mineral on or near OC surfaces, the key working minerals for C stabilization are essentially SRO minerals and/or poorly crystalline submicron size clay minerals. We have other sets of unpublished 3-D tomography data for samples from various mineralogy background, and the minerals found associated with OC

surface are coincidentally in line with nano particles/poorly crystalline minerals.

Inspired by the reviewer's suggestions, efforts have been taken to identify micro assemblage features which likely reflect the differentiation of adsorption and co-precipitation in our natural OC-mineral consortium using TXM images. In the all-depth 2-D X-ray absorption-contrast image (Fig. 1r), sheet-like mineral coating is observed on OC surface, which is a dense and thin layer, pointing to likely high level of physical protection. We propose such texture originate from absorption. Another distinct texture is recognized as OC-mineral cluster, with either minerals in the core and organic matter around (Fig. 1r), or vice versa (Fig. 2r). This type of texture indicates possible microsite of mineral co-precipitation with relatively free OC. Many clusters of various shapes are observed (Fig. 2r). However, to fully explore this long-standing question on the structural development of OC-Fe mineral assemblages attributed by absorption vs co-precipitation, we suggest future work with carefully planned lab-made OC-Fe consortiums and thorough 3-D tomography observation are needed for reliable evidence and answers.

Revised manuscript is attached (shown as a supplement file).

Please also note the supplement to this comment:
https://www.biogeosciences-discuss.net/bg-2017-509/bg-2017-509-AC2-supplement.pdf

––––––––––––––––––––––––––––––––

[Figure]

**Fig 1r. The 2-D X-ray absorption-contrast composite image in focus depth for OC-mineral consortium from Mt. Nanhua.** The grey scale is inverse proportional to X-ray attenuation coefficient of different materials. The dark laminal texture pointed by the white arrow reveals sheet-like mineral coating on OC surface, which is a dense and thin layer, pointing to likely high level of physical protection. We propose such texture originate from absorption. On the other hand, the red arrow points to the other distinct texture of OC-mineral cluster, with dark minerals in the core and light organic matter surrounding, which is not necessary on the rim. This texture indicates possible microsite of OC-mineral co-precipitation. It should be noted that the light region surrounding dark region (minerals) could be either organic matter or air/cavity, as their attenuation coefficient is very difficult to distinguish in X-ray image. There are numerous such OC-mineral clusters in the image, a large round-shape one was chosen for illustration. However, in reality, the OC-mineral cluster could be of any shape. The scale bar is 3 microns.

**Fig. 1.**

[Figure]

**Fig 2r. A Cross-section view of the reconstructed 3-D tomography**, under absorption contrast mode for the OC-mineral consortium from Mt. Nanhua, along the X-Z plane. The grey scale is proportional to X-ray attenuation coefficient of different materials. The white and the red arrows point to the mineral layer and OC-mineral cluster respectively. The yellow arrow points to a potential nucleus for mineral clusters development, it could be light material such organic C. On the other hand, this microsite may just be a cavity. The scale bar is 3 microns.

**Fig. 2.**

**Supplement:**

[revised manuscript text omitted]

Nano scale 3-D tomography revealed a high heterogeneity within OC-mineral consortium and most particulate OC surface is coated by minerals.  Natural OC exhibited strong organo-mineral association on its surface at nano scale in high mountain soil (Fig. 4; Fig. SMOV3). Abundant short-range order minerals in forms of subhedral particle or anhedral nano-aggregate have direct association with the boundary of OC, and develop coating on the tracheid surface (Fig. 4 b and c) (Mikutta et al., 2006). Sheet-like mineral coating is observed on OC surface, which are dense and thin layers, likely originated from absorption. Another distinct texture is recognized as nano-aggregates/clusters of various shapes, possibly formed by OC-mineral co-precipitation. Mineral aggregation by poorly crystalline nano particles renders natural sub-micron porosity, which may contribute to elevated sorption capacity in soil (Rawal et al., 2016). The densely-packed mineral texture suggested significant physical protection on OC surface (Kaiser and Guggenberger, 2007). The sorbed minerals not only can form physical protection, but also could shield OC from chemical weathering (Mikutta et al., 2006). The key working minerals for OC-mineral interplay are essentially nano particles/submicron/SRO/poorly crystalline minerals.

The nature of associated minerals was confirmed to be mainly SRO Fe oxyhydroxides, specifically ferrihydrite (ICDD 01-073-8408), goethite (ICDD 01-073-6522), lepidocrocite (ICDD 00-044-1415), and quartz (ICDD 00-033-1161) (Fig. 5; Table S1), analyzed using high resolution synchrotron-based XRD. Quartz may be at most a minor component on OC surface, considering their chemistry and particle size. Yet siliceous mineral surfaces may become coated with a veneer of hydrous Al- and Feoxides, which could confer net positive charge and promote their reactivity in tropical environments (Chen et al., 2014a; Sposito, 1989).

Considering their high surface area and reactivity, the abundant nano scale Fe oxyhydroxides could play a significant role for OC long-term stabilization through chemical bonding and physical shielding (Eusterhues et al., 2005; Kaiser et al., 2002;

Kiem and Kogel-Knabner, 2002; Mikutta et al., 2006), as well as cation sorption in soil, and contribute to longevity of OC in high mountain soil. According to elemental mapping results, aluminosilicates may also be present, however, their portion and crystalline level should be low due to their minimal signal in the XRD spectra (Figs. 5, 6). The FTIR

analyses reveal the chemistry of organo-mineral association (Fig. 7; Table S2). The aged

OC is highly aromatic when at the same time highly reactive, as broad bands centered at

$cm^{-1}$ and 1706 $cm^{-1}$ for aromatic C=C stretching and $\nu$ C=O in carboxyl are observed , pointing to likely origin of pyrogenic C (Özçimen and Ersoy-Meriçboyu, 2010;

Sharma et al., 2004). Both aromatic and carboxyl C functional groups normally have high affinity with Fe (III) (Mikutta et al., 2007; Zhao et al., 2016). The broad bands point to possible significant degree of association between OC and minerals (Chen et al., 2016;

Gu et al., 1994; Kaiser and Guggenberger, 2007). The sorption of OC to Fe oxyhydroxides through organo-mineral multiple complex bonds such as 'ligand exchange' could occupy and consume the reactive surface sites on OC and Fe oxyhydroxides, tune down their activity and enhance their respective stabilization (Chorover and Amistadi, 2001; Cornell and Schwertmann, 2006; Hall et al., 2016; Kaiser and Guggenberger, 2007; Mikutta et al., 2007). The discovery of short-range-order mineral ferrihydrite in air-dried OC particles and later ground samples indirectly validates its stabilization due to organo-mineral interplay. As a SRO/metastable mineral, ferrihydrite is hard to estimate accurately in dry soil samples due to its transient nature and the limitation of traditional extraction and spectroscopic methods (Cornell and Schwertmann, 2006). The specific mineral forms phase in direct contact with OC on surface at nano scale warrants future study (Fig. SMOV3). In-situ mineral mapping of different SRO minerals/ Fe oxyhydroxide on OC surface will provide mechanistic evidence on OC stabilization. Mineral physical protection on OC may represent the end stage of carbon stabilization, especially in a weak leaching and weathering environment.

Our in-situ description of organo-mineral interplay at nano scale provides direct evidence on the important role of mineral physical protection for OC long term stabilization. High amounts of ferrihydrite and other Fe oxyhydroxides were also found associated with lignin-like OC in soil under an aquic moisture regime (Eusterhues et al., 2011). The abundance of mineral nano particles, and their high heterogeneity and short-range-order nature could be common in humid environment, however, they could have been seriously underestimated by traditional analysis methods, such as electron microscopy/X-ray diffraction/fractionation approaches, which focus on clay-size minerals (Mikutta et al., 2005). Mineral physical protection for OC stabilization may be more important than previous understanding. More research is proposed to explore: i) whether it is a general phenomenon that the minerals interact with OC are essentially nano particle/submicron/SRO/poorly crystalline minerals, ii) whether the major mineral for C stabilization is nano scale SRO instead of clay-size mineral in soils. Perspective on C stabilization and saturation should take account the role of SRO minerals into soil C dynamics modelling, besides clay type and content.

[revised manuscript text omitted]

---

## Author Response (AR2)

**Reply to the Comments of Associate Editor and two reviewers by Weng et al. (bg-2017-509)**

Dear Editor:

We thank the Associate Editor and the two reviewers for their constructive comments and suggestions. We've done a thorough revision to improve the scientific writing and readability of our manuscript. We have endeavored to respond to all the suggestions and comments to improve the clarity and potential impact of our manuscript. Our manuscript has been edited by a professional English-speaking editor. Detailed responses are given in the following sections. Please see the "marked version" of the revised manuscript for the details of the changes that were made. One figure was added to the supplementary information to clarify the mineralogy of the mountain soil sample (Fig. S4).

Sincerely, Biqing Liang Ph.D, on behalf of all co-authors

**Responses to Editor's comments:**

Comments: I would find relevant to insert in your article most of your discussions/arguments/auto-criticisms we find in your letter of rebuttal. That will valorize the work you've done in response to this provacating referee.

Reply: We thank the Associate Editor for this reminder. Our first rebuttal letter carried a significant amount of background information on the topics of C stabilization and saturation, and it was mainly prepared to align and update Reviewer #1's understanding on these topics and hope to fill in the knowledge gap. We've done a thorough revision to incorporate all relevant points of discussions and arguments into the revised manuscript.

Specifically, we rephrased our important results for clarity and added the paragraph into the abstract: "In situ evidence revealed an abundance of mineral nanoparticles, in dense thin layers or nano-aggregates/clusters, instead of crystalline clay-sized minerals on or near OC surfaces. The key working minerals for C stabilization were reactive short-range-order (SRO) mineral nanoparticles and poorly crystalline submicron-sized clay minerals. Spectroscopic analyses demonstrated that the studied OC was not merely in crisscross co-localization with reactive SRO minerals. There could be a significant degree of binding between OC and the minerals." Please see Page 2, Line 1-6.

We made a substantial revision on the first part of the introduction to articulate the inadequate understanding of SRO minerals in real soil environment and the imperative need of high resolution 3-D tomography for probing nano-sized minerals in association with OC at nanometer scale. Please see Page 2, Line 18-21, and Page 3, Line 1-16. We extended knowledge on the application of X-ray tomography on soil and organic C samples and highlighted the unique strength of our approach. Please see Page 4, Line 5-10; Page 6, Line 3-11.

We also strengthened our results and conclusions. Please see Page 13, Line 9-17 and Line 20-22; Page 14, Line 1-5; Page 14, Line 19-22; Page 15, Line 1-2, Line 10-11, and Line 18-20; Page 16, Line 6-22; Page 17, Line 1-8. Based on our finding, we recommended future research topics for the scientific community and suggested the need to incorporate SRO reactive minerals and BC/pyrogenic C into soil modeling: "We recommend future research to: (1) explore whether it is a general phenomenon that the minerals interacting with OC surface are mineral nanoparticles and submicron-sized clay minerals, and (2) explore whether the mineral for C stabilization is primarily nano-sized SRO minerals instead of clay-sized minerals in soils. Perspectives on C stabilization and saturation may be revolutionized once the role of SRO minerals is considered in modeling soil C dynamics, in addition to parameters such as clay type and content. We suggest that the modeling of SOC turnover should also include BC/pyrogenic OC into the biochemically protected pool as BC/pyrogenic OC can persist over millennials under natural exposure." Please see Page 16, Line 9-17.

Comments: You did not remove from your discussion this poor sentence 'Mineral physical protection for OC stabilization may be more important than previous understanding'. This sentence can be easily replaced by some sentences of your letter of rebuttal.
Reply: Thank you for the reminder. We deleted the sentence "Mineral physical protection for OC stabilization may be more important than previous understanding" and also modified the sentences immediately preceding it. Please see Page 16, Line 6-9.

Comments: I would also ask you to correct your manuscript by a native English speaker. For example, the sentence 'i) whether it is a general phenomenon that the minerals interact with OC are essentially SRO mineral nano particles and or poorly crystalline submicron-sized clay minerals' has no clear meaning.
Reply: Our manuscript has been edited by a professional English-speaking editor. The sentences were rephrased as "We recommend future research to: (1) explore whether it is a general phenomenon that the minerals interacting with OC surface are mineral nanoparticles and submicron-sized clay minerals, and (2) explore whether the mineral for C stabilization is primarily nano-sized SRO minerals instead of clay-sized minerals in soils.

**Reply to Reviewer#1**

Comments: In this paper, Weng et al. study the 3D distribution of organic carbon and mineral particle using synchrotron-based transmission X-ray microscopy. The approach was applied to a lab-made mixture of black carbon and nano mineral ($TiO_2$) and to a natural soils rich Fe oxyhydroxides.

Reply: We've developed a high resolution 3-D tomography approach and successfully applied it to study the in situ interplay of organic C and associated minerals in a lab-made and natural OC-mineral consortium at nanoscale. We discovered that the stabilization of the 3500-year-old natural OC was mainly attributed to the physical protection of nano-sized Fe-containing mineral (Fe oxyhydroxides) and to the strong organo-mineral complexation. We provided in situ evidence and revealed an abundance of mineral nanoparticles, in dense thin layers or nano-aggregates/clusters, instead of crystalline clay-sized minerals on or near OC surfaces. The key working minerals for C stabilization were reactive short-range-order (SRO) mineral nanoparticles and poorly crystalline submicron-sized clay minerals. According to the XRD spectrum, the primary minerals in the mountain soil were quartz, amesite (Kaolin-serpentine), and muscovite (Fig. S4). In addition, minimum Fe oxyhydroxides signal was observed in the mountain soil, indicating a Fe oxyhydroxides-poor environment. One more figure of soil mineralogy compared to the particulate OC was added into the supplementary information for clarification (Fig. S4).

Comments: By observing crisscross of mineral particles and organic matter they conclude on the importance of mineral physical protection (for example through ligand exchanges) for long-term persistence of OC.

Reply: Spectroscopic analyses demonstrated that the studied OC was not merely in crisscross co-localization with reactive SRO minerals. There could be a significant degree of binding between OC and the minerals. The FTIR analyses demonstrated the chemistry of organo-mineral association (Fig. 7; Table S2). The aged OC is highly aromatic when it is highly reactive. Broad bands are observed at 1596 cm$^{-1}$ for aromatic C=C stretching and 1706 cm$^{-1}$ for carboxylic ν C=O. Both aromatic and carboxyl C functional groups have high affinity with Fe (III) (Hall et al., 2016; Mikutta et al., 2007; Zhao et al., 2016). Broad bands indicate a significant degree of association between OC and minerals (Chen et al., 2016; Gu et al., 1994; Kaiser and Guggenberger, 2007). Please see Page 2, Line 1-6; Page 15, Line 2-5. Ligand exchange is within the organomineral multiple complex bonds which have been proposed for C stabilization. However, the specific bonding type was not validated in situ. Thus, we deleted "such as ligand exchange" and rephrased the sentence in Page 15, Line 10-11.

Comments: They also conclude that mineral physical protection for OC stabilization may be more important than previously thought.

Reply: We did not intend to make this statement as a conclusion. Rather, it was a description of our perspective. To avoid contention, we deleted this sentence from the abstract and the main text. Please see Page 2, Line 10-11; Page 16, 8-9.

Comments: I have the feeling that this study uses a Jack Hammer to open nuts.

Reply: We made scientific contribution to develop a high resolution synchrotron-based 3-D tomography approach at nanometer scale, which is a promising tool for generating new insight into the interior 3-D structure of micro-aggregates, the in situ interplay between OC and minerals, and the fate of mineral nanoparticles (including heavy metals) in natural environments.

We must point out that in real soil environments, the actual key mineral players (crystalline clay mostly in micron-sized vs nanoscale SRO minerals) for in situ interaction with OC have not been identified, and their spatial and temporal variations are unknown. To date, no information is available on the in situ distribution of reactive minerals below clay size, or on the subsequent organo-mineral micro-assemblage in soils (Page 3, Line 8-16).

We advocate scientific efforts based on in situ evidence and not merely based on general ideas or concepts. Non-destructive high-resolution X-ray 3-D tomographic technique is vital for the study of interfacial minerals on OC surface, and for in situ evidence. The approach we developed using dual-scan modes overcomes (1) the limitation of bulk sample posed by traditional fractionation method (Page 3, Line 18-20), (2) the limitation of spatial resolution by micron-CT and non-synchrotron-based 3-D tomography (Page 4, Line 5-10; Page 5, Line 3-11), and (3) the limitation of poor X-ray absorption in light materials such as organic C (Page 5, Line 5-11). The approach succeeded in revealing the size, particle shape, and distribution of the interfacial minerals on OC surface at nanoscale. We may further link the phase, level of crystallinity, and reactivity of minerals to their active role of OC stabilization.

High resolution X-ray scanning and 3-D tomography is highly demanding in terms of technology, multi-disciplinarity and big-data analysis (Lafond et al., 2015). Except for the adoption of high resolution X-ray objective lens, the fidelity of 3-D tomography relies on the accurate alignment of the 2-D projections in correct three-dimensional positions. However, non-negligible mechanical imperfection of the rotational stages at nanometer level, or the thermal effects may significantly degrade the spatial resolution of reconstructed tomography. We have developed a markless image auto-alignment algorithm for fast projection matching (**Faproma**, Wang et al., 2017) to overcome these challenges, and accomplished accurate reconstruction of 3-D tomography at the nanometer level. Please see Page 6, Line 3-11.

Comments: We already know that minerals play key role in the persistence of organic matter in soils. The specific role of reactive minerals such as Al and Fe oxyhydroxides, allophane is known since several decades. Therefore, I do not see what progress in our knowledge is made here.

Reply: The role of mineral, especially reactive minerals on C stabilization are well recognized, mostly based on their correlation in the extracted fractions. For example, the accumulation and subsequent loss of OC have been found largely driven by changes in the millennial scale cycling of mineral-stabilized C, and a positive correlation between non-crystalline minerals and OC has been found in soils across a climate gradient (Torn et al., 1997). Moreover, SRO minerals, which may only exist in small portions, and mostly in the fine and dense soil fraction, are crucial for C stabilization (Cusack et al., 2012; Eusterhues et al., 2005; Kaiser et al., 2002a; Mikutta et al., 2005; Mikutta et al., 2006; Mikutta et al., 2007; Rasmussen et al., 2005). Please see Page 3, Line 2-6.

Even though SRO minerals are known for high reactivity, their development of strong organo-mineral complexation with OC relies on immediate contact with the reactive surface of OC. On the other hand, although SRO normally accounts for only a small portion of weight in the soil fraction, their distribution could be prevalent due to small size. To date, no information is available on the in situ distribution of reactive minerals below clay size in real soil environment. Our in situ evidence revealed an abundance of mineral nanoparticles, in dense thin layers or nano-aggregates/clusters, instead of crystalline clay-sized minerals on or near OC surfaces in natural OC-mineral consortium. The key working minerals for C stabilization were reactive short-range-order (SRO) mineral nanoparticles and poorly crystalline submicron-sized clay minerals in the studied mountain soil. Our research triggered two new research directions for the scientific community: (1) to explore whether it is a general phenomenon that the minerals interacting with OC surface are mineral nanoparticles and submicron-sized clay minerals, and (2) to explore whether the mineral for C stabilization is primarily nano-sized SRO minerals instead of clay-sized minerals in soils." Perspectives on C stabilization and saturation may be revolutionized once the role of SRO minerals is considered in modeling soil C dynamics, in addition to parameters such as clay type and content. We suggest that the modeling of SOC

turnover should also include BC and pyrogenic OC into the biochemically protected pool as BC can persist over millennials under natural exposure. Please see Page 16, Line 9-17.

Comments: Moreover, the observation of crisscross of mineral particles and organic matter in one soil layer does not inform on the role of minerals on OM persistence and does not permit any generalization. We do not know whether the OM is really retains by minerals, with what forces and what consequences for its fate in soils.

Reply: Spectroscopic analyses demonstrated that the studied OC was not merely in crisscross co-localization with reactive SRO minerals. There could be a significant degree of binding between OC and the minerals. We have explained this point in details in the second point of reply. We concluded that the stabilization of the 3500-year-old natural OC was mainly attributed to the physical protection of nano-sized Fe-containing minerals (Fe oxyhydroxides) and to the strong organo-mineral complexation in the mountain soil we sampled. We did not intend to use only one soil profile for a generalization of conclusion on the mineral stabilization of OC. It is worth noting that, according to XRD spectrum, the primary minerals in the soil were quartz, amesite (Kaolin-serpentine), and muscovite (Fig. S4). In addition, minimum Fe oxyhydroxides signal was observed. The soil was not rich in Fe oxyhydroxides; however, the particulate organic C studied was rich in Fe oxyhydroxides. The key working minerals for physical protection and chemical stabilization of OC could be nano-sized SRO minerals in a soil that is not rich in Fe oxyhydroxides. Our results suggest:

i)      The coated particulate OC surfaces by SRO mineral nanoparticles are stabilized.

ii)     Minerals physical protection contributes to the long term persistence of OC (3500-year-old) in the sub-tropical environment.

iii)    Substantial number of mineral surfaces are likely available for C stabilization.

Comments: From my modest knowledge, I think that what we need now is to determine whether the capacity of minerals to fix carbon is limited; If this capacity is limited, how much carbon can still be stored in soils (by considering the whole soil profile). Understanding how some organic compounds free of minerals can persist over centuries (An example among many others: Derenne et al. 1991) could be another original and exciting research avenue.

Reply: The notion of carbon saturation is developed on the assumption that mineral physical protection could be limited by soil physiochemical characteristics such as silt and/or clay content, microaggregate, and surface area (Hassink, 1997; Kemper and Koch, 1966). However, little is known about how spatial and temporal variation in soil mineralogy (especially type, phase, and reactivity) controls C long-term stabilization.

To determine whether the capacity of minerals to fix carbon is limited or not, Six et al. (2004) suggested that quantifying the protective capacity of a soil requires a careful consideration of all mechanisms of protection and the implications of experimental procedures. They also advocated a multi-dimensional view of the soil heterogeneity. When we intend to estimate the C fixation capacity of soil and how much carbon can still be stored, it is important for us to first get the right and solid idea about the real soil environment. A sound understanding of the three-dimensional interplay of organic C and minerals requires in situ information at the microaggregate level, especially on the different minerals' phase, size, distribution, and spatial co-localization with organic C at the nanoscale. For example, a substantial parts of mineral surfaces are considered likely not covered by organic C (Arnarson and Keil 2001; Kahle et al. 2002; Mayer and Xing 2001; Ransom et al. 1998). New perspectives call for detailed observational evidence on OC-mineral association and micro-assemblage at nanoscale. We unveil:

i)      There is a high heterogeneity within OC-mineral consortium, including many nano/submicron-sized mineral particles, microsites and extensive porosity.

ii)     Substantial number of mineral surfaces are likely available for C stabilization.

We studied black carbon and natural pyrogenic C which can persist in natural environment over millennia and should fall into the scope of original and exciting research as suggested by Reviewer #1. Six et al. (2002) pointed out that biochemically stabilized organic compounds which could persist in the environment for centennials to millennials normally are partially stabilized by association with minerals (e.g. clay and silt particles).

**Response to Reviewer#2**:

Comments: This study aimed to develop the 3-D tomography of organic carbon-mineral consortium using the synchrotron based transmission X-ray microscopy. Both lab-made and natural black carbon were tested. Results of in-situ 3-D tomography directly indicate the association between OC and minerals in both samples. For the natural black carbon, other spectroscopic results including XRD and FTIR also demonstrate the abundance of Fe (hydr)oxides and their significance in the physical stabilization of OC. Although the association between Fe minerals and OC has been well known for decades, this study showed the direct and clear evidence for such mechanism. The contribution of this study lies in the fact that authors developed the methodology to obtain the 3-D images using the TXM. In terms of the C sequestration, the degree of C stabilization is related to the structural development of Fe/C assemblages. Previous studies could only gauge the adsorption vs. co-precipitation between OC and Fe minerals via the indirect spectroscopic evidences. The 3-D technique developed here provides a new insight to determine the structure of such micro-aggregates.

Reply: We thank Reviewer #2 for recognizing our work in methodology development and contribution to the understanding of C stabilization.

Comments: In this case, I would suggest authors to add more information regarding the differentiation of adsorption vs. co-precipitation between OC and Fe minerals using the 3-D tomography results to shed light on its significance toward the C stabilization.

Reply: Inspired by reviewer's suggestions, efforts have been taken to identify micro assemblage features which indicate adsorption or co-precipitation in our natural OC-mineral consortium using TXM images. In the all-depth 2-D X-ray absorption-contrast image (Fig. S2), a sheet-like mineral coating is observed on OC surface, which is a dense and thin layer, likely indicating a high level of physical protection. We propose such texture originates from adsorption. Another distinct texture is recognized as OC-mineral nano-aggregates/clusters, with either minerals in the core and OC around (Fig. S2), or vice versa (Fig. S3). This type of texture indicates possible OC-mineral co-precipitation at microsites. Many clusters of various shapes are observed (Fig. S3). However, to fully explore this long-standing question on the structural development of OC-Fe mineral assemblages attributed by adsorption vs co-precipitation, we suggest that further research with carefully designed samples (such as distinct lab-made OC-Fe consortiums) and enough 3-D tomography observations are needed for reliable evidence and answers.

We have added Figures S2 and S3 into the supplementary information, please review for details, thank you.

[revised manuscript text omitted]
 the measured distributions, and 3-D tomography illustration is generated by the image post-process and computation.

[Figure]

**Figure S2.** The 2-D X-ray absorption-contrast composite image in all focus depths for OC-mineral consortium from Mt. Nanhua. The grey scale is proportional to the X-ray attenuation coefficients of different materials. The dark laminal texture pointed by the white arrow reveals a sheet-like mineral coating on OC surface, which is a dense and thin layer, likely indicating a high level of physical protection. We propose such texture originates from adsorption. The red arrow points to the other distinct texture of OC-mineral nano-aggregates/clusters, with dark minerals in the core and light OC surrounding, which is not necessarily on the rim. This texture indicates a possible microsite of OC-mineral co-precipitation. It should be noted that the light region surrounding the dark region (minerals) could be either OC or air/cavity, as their attenuation coefficients are very difficult to distinguish from each other in X-ray images. There are numerous such OC-mineral clusters in the image, with a large, round-shaped one chosen for illustration purposes. In reality, the OC-mineral clusters can be of any shape. The scale bar is 3 microns.

[Figure]

**Figure S3.** A cross-sectional view of the reconstructed 3-D tomography, under absorption-contrast mode for the OC-mineral consortium from Mt. Nanhua, along the

X-Z plane. The grey scale is inversely proportional to the X-ray attenuation coefficients of different materials. The white and the red arrows point to the mineral layer and OC- mineral clusters respectively. The yellow arrow points to a potential nucleus for mineral clusters development that could be a light material such as OC. On the other hand, this microsite may be a cavity. The scale bar is 3 microns.

[Figure]

**Figure S4.** The X-ray diffraction patterns of the particulate OC and the mountain soil.

There is a distinct difference between the particulate OC and the mountain soil in mineralogy as shown in the stacking graph. The major crystal phases in the soil is quartz (ICDD 0-033-1161), amesite (ICDD 01-080-1772), and muscovite (ICDD 04-012-

1956). The X-ray diffraction pattern also shows a few minor phases in the mountain soil, such as chlorite (ICDD 01-075-8791), phlogopite (ICDD 00-016-0344), magnetite (ICDD 01-080-6407), and hematite (ICDD 01-080-5408), but these diffraction peaks are hardly recognizable due to their low intensity.

[Figure]

[Figure]

**Figure SMOV1.** Video illustration extracted from 3-D absorption-contrast tomography of lab-made BC and mineral nanoparticle consortium. The yellow particle is a gold nanoparticle for position reference. All minerals are shown in a silver color. The dark grey part contours the structure and boundary of OC.

https://drive.google.com/open?id=1FD-ui0-lsr4U2eClo6X2AbwqtcuChtII

**Figure SMOV2.** Video illustration extracted from 3-D phase-contrast tomography of lab-made BC and mineral nanoparticle consortium. The yellow particle is a gold nanoparticle for position reference. All minerals are shown in a silver color. The dark grey part contours the structure and boundary of OC.

https://drive.google.com/open?id=1RglvAplyXrnZTIZQyr7aGTo8vYGbkCJu

[Figure]

**Figure SMOV3.** Video illustration obtained from 3-D absorption-contrast tomography of the particulate mineral-bearing OC from the mountain soil. The yellow particle is a gold nanoparticle for position reference. All minerals are shown in a rust color. The dark grey part contours the structure and boundary of OC.

https://drive.google.com/open?id=1-__9KHc3SpncXfuflMy9lQ8V0AIVmB8d

Table S1. XRD peak positions of mineral-bearing OC sample from Mt. Nanhua.

| | d (Å) | d-reference (Å) | hkl |
|---|---|---|---|
| Ferrihydrite | 2.5644 | 2.5634 | 100 |
| | 2.2502 | 2.2504 | 012 |
| | 2.0046 | 1.9840 | 013 |
| | 1.7344 | 1.7322 | 014 |
| | 1.5090 | 1.5160 | 015 |
| | 1.4779 | 1.4800 | 110 |
| Goethite | 4.9831 | 5.0000 | 020 |
| | 4.2063 | 4.2089 | 110 |
| | 2.6992 | 2.7071 | 130 |
| | 2.5914 | 2.5913 | 021 |
| | 2.4595 | 2.4591 | 111 |
| | 2.2625 | 2.2624 | 121 |
| | 1.7210 | 1.7284 | 221 |
| | 1.6990 | 1.7005 | 240 |
| | 1.5650 | 1.5706 | 151 |
| | 1.5135 | 1.5150 | 002 |
| Lepidocrocite | 6.2651 | 6.2700 | 200 |
| | 3.2921 | 3.2940 | 210 |
| | 2.4747 | 2.4730 | 301 |
| | 2.4333 | 2.4340 | 410 |
| | 2.3616 | 2.3620 | 111 |
| | 1.9402 | 1.9400 | 501 |
| | 1.9370 | 1.9350 | 020 |
| | 1.7367 | 1.7350 | 511 |
| | 1.5333 | 1.5340 | 002 |
| | 1.5258 | 1.5240 | 321 |
| | 1.3684 | 1.3710 | 521 |
| Quartz | 4.2532 | 4.254 | 100 |
| | 3.3422 | 3.342 | 101 |
| | 2.4571 | 2.456 | 110 |
| | 2.2806 | 2.280 | 102 |
| | 2.2361 | 2.236 | 111 |
| | 1.9788 | 1.979 | 201 |
| | 1.8173 | 1.817 | 112 |
| | 1.6715 | 1.671 | 202 |

|        |        |     |
|--------|--------|-----|
| 1.5412 | 1.541  | 211 |
| 1.3818 | 1.374  | 203 |

**Table S2.** FTIR peak assignment of mineral-bearing OC sample from Mt. Nanhua.

| Wavenumber (cm$^{-1}$) | Model | Reference | Ref. value |
|---|---|---|---|
| 1758 | Carbonyl C=O stretching | Parikh et al., 2014 | 1765 |
| 1706 | Aromatic carbonyl/carboxyl C=O stretching | Özçimen and Ersoy-Meriçboyu, 2010 | 1709 |
| 1596 | vC=C in aromatic | Sharma et al., 2004 | 1597 |
| 1454 | CH deformation and aromatic ring vibrations | Sharma et al., 2004 | 1460 |
| 1386 | Carboxyl C–O symmetric stretching | Parikh et al., 2014 | 1384 |
| 1274 | Carboxyl C–O stretching | Parikh et al., 2014 | 1280 |
| 1247 | v(C-O) phenolic | Parikh et al., 2014 | 1240 |
| 1113 | Si–O stretching | Vaculikova et al., 2011 | 1113 |
| 1062 | Si–O stretching | Harsh et al., 2002 | 1060 |
| 1025 | Aliphatic ether C–O and alcohol C–O stretching | Parikh et al., 2014 | 1029 |
| 910 | OH deformation | Vaculikova et al., 2011 | 913 |
| 875 | 1 adjacent H deformation | Parikh et al., 2014 | 870 |
| 798 | 2 adjacent H deformation | Parikh et al., 2014 | 804 |
| 754 | 4 adjacent H deformation | Parikh et al., 2014 | 750 |
| 694 | Fe-OH stretching | Blanch et al. 2008 | 690 |
| 674 | In-plane O-H bend | Blanch et al. 2008 | 670 |
| 626 | Fe–O stretching | Blanch et al. 2008 | 633 |
| 534 | Fe-OH stretching | Blanch et al. 2008 | 533 |
| 497 | Fe–O asymmetric stretching | Blanch et al. 2008 | 497 |
| 476 | Fe-O vibrations | Parikh et al., 2014 | 480 |

(Blanch et al., 2008; Harsh et al., 2002; Özçimen and Ersoy-Meriçboyu, 2010; Parikh et al., 2014; Sharma et al., 2004; Vaculíková et al., 2011)